# Validation of a Methodology for the Quantification of DON in Feces and Feedstuffs by UPLC as Possible Strategy to Evaluate the Detoxifying Efficacy of a Mycotoxin Adsorbent In Vivo

**DOI:** 10.3390/toxins17070322

**Published:** 2025-06-24

**Authors:** Bo Yang, Hui Deng, Yiwei Jia, Dong Li, Rudeng Chen, Ruiqing Chen, Jing Zhang, Yan Zhong, Lingxian Yi, Fuhao Wang, Hongjie Cui, Daojin Yu

**Affiliations:** 1University Key Laboratory for Integrated Chinese Traditional and Western Veterinary Medicine and Animal Healthcare in Fujian Province/Fujian Key Laboratory of Traditional Chinese Veterinary Medicine and Animal Health, College of Animal Sciences, Fujian Agriculture and Forestry University, Fuzhou 350002, China; ybvet@fafu.edu.cn (B.Y.); deng12345hui@163.com (H.D.); j1937585867@163.com (Y.J.); zhongyan2506@163.com (Y.Z.); lingxian_yi@outlook.com (L.Y.); wfh4399@gmail.com (F.W.); 2Wuhan Animal Disease Control Center, Wuhan 430016, China; dongli0905@126.com; 3Fujian Inspection and Testing Centre for Agricultural Product Quality and Safety, Fuzhou 350003, China; chenrudeng1@gmail.com (R.C.); chenruiqing215@gmail.com (R.C.); j200712263@163.com (J.Z.)

**Keywords:** deoxynivalenol, ultra-performance liquid chromatography, in vivo evaluation of detoxification efficacy of mycotoxin adsorbent, fecal sample

## Abstract

The study aimed to provide a possible strategy to evaluate the detoxifying efficacy of mycotoxin adsorbents in vivo by analyzing deoxynivalenol (DON) concentration in feces. Fifteen pigs were randomly assigned to five groups (groups A–E, 3 replicates/group). The pigs in each group were fed twice a day for 10 d with 500 g of designed diets (group A, commercial feedstuffs; group B, DON-contaminated (mildewed) feedstuffs; groups C, D, E, mildewed feedstuffs containing 0.2% adsorbent 1, 2, and 3, respectively). For each pig, 2-g fecal samples were collected pre-feeding and analyzed by ultra-performance liquid chromatography. Nondetectable or low concentrations of DON (<1.38 μg/g) were found in fecal samples from groups A and B. High concentrations of DON (>20 μg/g) were detected in six out of twenty fecal batches from pigs in group C. Moderate concentrations of DON (5.54–6.50 μg/g) were detected in one out of twenty fecal batches from pigs in group D and two out of twenty in group E. Based on the predefined evaluation criteria, higher DON concentration and frequency in feces indicate better adsorbent efficacy. Notably, Absorbent 1 demonstrated a more pronounced detoxification efficacy in vivo compared to the other two absorbents.

## 1. Introduction

Deoxynivalenol (DON) is a secondary metabolite produced by *Fusarium* species. It is considered the most prevalent mycotoxin in feedstuffs [1,2,3]. The contamination of feedstuffs with DON can cause huge economic loss to the livestock industry. The consumption of DON-contaminated feedstuffs causes loss of appetite, vomiting, gastroenteritis, reduced weight gain and reproductive capacities, immune dysfunction, or even death in farm animals [4,5], while promoting the absorption of DON itself in the intestine lumen and increasing its toxicity [6]. Pigs are the most sensitive farm animals to DON due to their relatively high oral bioavailability (>55%), absence of detoxification intestinal microorganisms, and long clearance time (2 h < t_1/2_ < 4 h) [7].

Many attempts have been made to prevent DON from contaminating feedstuffs or to detoxify DON-contaminated feedstuffs. However, the contamination of feedstuffs with DON is difficult to avoid, even under good agricultural practices, storage, and transportation conditions. Adding mycotoxin adsorbents to DON-contaminated feedstuffs may be a convenient and effective method. DON in feedstuffs can be adsorbed on the surface of mycotoxin adsorbents. In the bound form, DON is difficult to be absorbed into the blood and is excreted in the feces [8]. Based on this principle, the toxic effects of DON on pigs are minimized.

So far, several mycotoxin adsorbents are commercially available, including inorganic adsorbents (e.g., clays, zeolite, and activated carbon) and organic adsorbents (e.g., yeast cell walls and micro-ionized fibers). These adsorbents bind mycotoxins through physico-chemical interactions such as ion-dipole forces, Van der Waals forces, and hydrogen bonding [9]. Regarding the evaluation methods for the detoxification efficacies of these adsorbents, each exhibits distinct advantages and disadvantages. Several in vitro methods have been reported for determining the adsorption capacities of some inorganic adsorbents [10,11,12,13,14,15,16,17,18], organic adsorbents [11,13,17,19,20,21,22,23], and commercial adsorbents [16,24,25]. These methods include single in vitro tests [10,11,13,14,15,16,18,19,23], adsorption isotherm tests [12,17,20,21], in vitro rumen models [22,25], and dynamic gastrointestinal models [24]. Despite the widespread use of in vitro methods, they still have many limitations, such as high variability in results [9]. Only a few studies have reported in vivo methods to evaluate the detoxification efficacies of mycotoxin adsorbents [23,26,27]. These methods evaluate the detoxification efficacies based on non-specific parameters such as growth performance, hepatic function, and inflammatory status [26], or the toxicokinetic profiles of mycotoxins and their metabolites in the body [23,27]. However, these non-specific parameters are not sufficient to prove the detoxification efficacies of mycotoxin adsorbents, although they are useful [28], and obtaining the toxicokinetic profiles of mycotoxins and their metabolites is difficult and time-consuming. More simple and effective in vivo evaluation methods are therefore required to promote the development of mycotoxin adsorbents.

According to the European Food Safety Authority (EFSA), only specific biomarkers for exposure, such as mycotoxins and their metabolites in feces, urine, and blood, can prove the detoxification efficacies of mycotoxin adsorbents [28]. When DON and mycotoxin adsorbents are orally ingested by pigs, the concentrations of DON in pig feces can better reflect the adsorption capacities (detoxification efficacies) of the adsorbents in vivo. However, few analytical methods are available for determining DON in fecal samples [29,30,31,32,33,34,35,36,37,38]. Most of these analytical methods were liquid chromatography-mass spectrometry (LC-MS). Ultra-performance liquid chromatography (UPLC) has lower instrument requirements and analysis costs than LC-MS and is more suitable for evaluating the detoxification efficacies of mycotoxin adsorbents in vivo.

Based on these considerations, we developed a UPLC method for the quantification of DON in feces and feedstuffs and investigated the concentrations of DON in pig feces after adsorbent administration.

## 2. Results

### 2.1. Performance of the UPLC Method

Representative chromatograms obtained from blank fecal and feedstuff samples, blank samples spiked with DON, and incurred samples collected after treatment were presented in Figure 1. The retention time of DON was approximately 12.3 min, and no interfering peaks from endogenous compounds were found around the retention time of DON. The standard curve showed excellent linearity (y = 0.4472x + 0.008, r = 1) over the concentration range from 0.02 to 50 μg/mL. The LOD and LOQ were 0.02 μg/g and 0.07 μg/g for fecal samples and 0.05 μg/g and 0.17 μg/g for feedstuff samples, respectively. The precision and accuracy of the UPLC method were summarized in Table 1. The intra-day and inter-day RSDs were less than 1.7% and 1.2% for fecal samples and 14.6% and 11.7% for feedstuff samples, respectively. The recoveries ranged from 84.4% to106.8% at spiking levels of 0.1, 1, and 10 μg/g in pig feces and 77.6% to 119.5% at spiking levels of 0.01, 0.1, and 1 μg/g in pig feedstuffs, respectively. The stability evaluation showed that DON was stable in pig feces and pig feedstuffs at −2°C for 7 d and at 60 °C for 2 d (Table 2).

### 2.2. Detoxification Efficacies of Tree Commercial Adsorbents

The concentrations of DON in commercial and mildewed feedstuffs during the preparation of mildewed feedstuffs and the animal experiment are shown in Figure 2. The results indicated that the highest DON concentration (approximately 2.05 μg/g) was attained when the commercial feedstuffs were processed as described above for 9 d. Additionally, the concentrations of DON in commercial and mildewed feedstuffs remained relatively stable during the animal experiment, showing no significant changes. On day 10 of the animal experiment, the pigs fed commercial feedstuffs (group A) showed increased (*p* < 0.05) body weight compared with the pigs fed mildewed feedstuffs (groups B, C, D, E), and the pigs fed mildewed feedstuffs showed varying degrees of clinical manifestations, characterized by rough and dull haircoats, prominent tear stains, depressive demeanor, and reduced feed intake. The concentrations of DON in pig feces throughout the animal experiment are shown in Table 3. High concentrations of DON (>20 μg/g) were detected in six out of twenty fecal batches from pigs in group C (fed mildewed feedstuffs and absorbent (1). Moderate concentrations of DON (5.54–6.50 μg/g) were detected in one out of twenty fecal batches from pigs in group D (fed mildewed feedstuffs and absorbent (2), and two out of twenty in group E (fed mildewed feedstuffs and absorbent (3). While regarding the fecal samples collected from the pigs in groups A and B, no DON was detected, or only a low concentration of DON (<1.38 μg/g) was found. Absorbent 1 demonstrated a more pronounced detoxification efficacy in vivo compared to the other two absorbents.

The concentrations of DON in feedstuff samples from six pig farms were determined as follows: farm 1 (1.55 ± 0.32 μg/g), farm 2 (2.74 ± 0.58 μg/g), farm 3 (1.89 ± 0.49 μg/g), farm 4 (0.87 ± 0.26 μg/g), farm 5 (3.24 ± 0.27 μg/g), and farm 6 (1.21 ± 0.33 μg/g). The concentrations of DON in pig feces collected from six pig farms are shown in Table 4. Among the 16 fecal samples collected from pig farms 1–3, where adsorbent 1 was used, 4 samples were detected with medium or high concentrations of DON. However, among the 15 fecal samples collected from pig farms 4–6, where adsorbent 2 was used, DON was not detected, or only low concentrations of DON were detected. The results of the adsorbent application trials conducted in six pig farms were basically consistent with those of our animal experiment.

## 3. Discussion

The widespread contamination of feedstuffs by DON poses significant challenges to the pig industry. Highly efficient mycotoxin adsorbents are considered an effective tool for addressing this challenge, making the scientific evaluation of their detoxification efficacies of great importance. In this study, a simple and reliable UPLC method was developed and validated for the quantification of DON in feces and feedstuffs. This method, combined with appropriate animal experiments, may provide a possible strategy to evaluate the detoxifying efficacies of three commercial mycotoxin adsorbents in vivo.

The detoxification efficacy evaluation of mycotoxin adsorbent using our strategy relies on quantifying the concentrations of DON in pig feces. All concentration data, including those from fecal and feedstuff samples, were derived from the animal experiment using our UPLC method. This analysis method was well validated as described above. The sample pretreatment procedure was optimized to enhance the analytical performance of the UPLC method. Before solvent extraction, fecal and feedstuff samples underwent drying, grinding, and sieving processes. This was conducted to ensure the homogeneous distribution of analyte (DON) in the samples, and the potential detection errors due to small sample size (150–300 mg) could thus be reduced. Additionally, the grinding and sieving procedures enhanced the interaction between the analyte and the extraction solvent. This, in turn, led to an improvement in the extraction recovery. Fecal and feedstuff samples contain large amounts of endogenous interferences. Some of these interferences can be co-extracted together with the analyte and enter the chromatographic system. This will cause poor analytical specificity and damage the chromatographic column. Reducing sample size and screening selective extraction solvents can help reduce the amount of co-extracted interferences in samples to be analyzed. During the development of UPLC method, a significant optimization was conducted on the sample size. Specifically, it was reduced from 5.0 g to a mere 150 mg. A comprehensive screening of extraction solvents was also conducted. Ten different solvents were considered for this purpose, including acetonitrile, methanol, ethyl acetate, ethanol, acetone, 70% methanol solution (*v*/*v*), 84% acetonitrile solution (*v*/*v*), isopropyl alcohol, dichloromethane, and dimethyl sulfoxide. Optimized extraction of DON from 150 mg fecal samples (or 300 mg feedstuff samples) with 1.5 mL acetonitrile yielded maximal extraction recovery while restricting co-extracted interferences to acceptable levels.

DON in feces may bind to the adsorbents added in feedstuffs, forming DON–adsorbent complexes. During the analysis of fecal samples, if the extraction solvent fails to efficiently separate DON from the DON–adsorbent complex, false negative results may occur. This, in turn, may cause misjudgment of the detoxifying efficacies of mycotoxin adsorbents (underestimating their detoxification efficacies). To evaluate the efficiency of acetonitrile for separating DON from the DON–adsorbent complexes, an appropriate amount of DON was spiked into three commercial adsorbents (150 mg each) to achieve a final concentration of 5 μg/g. The mixtures were homogenized by thorough stirring and equilibrated at room temperature for 24 h, with 5 replicate samples prepared for each adsorbent. Subsequently, DON was extracted with acetonitrile, and the extracts were analyzed by UPLC. The extraction procedures and chromatographic analysis were identical to those in our UPLC method. The results showed that acetonitrile effectively separated DON from the complex, yielding a recovery rate of 98.3 ± 3.2%.

The mobile phase composition was also optimized since it could affect the chromatographic separation and analytical sensitivity. In previously published HPLC methods, DON was eluted from the chromatographic column with methanol–water, acetonitrile–water, and methanol–acetonitrile–water mixtures [29,30,31]. The content of organic solvent (methanol or acetonitrile) in the mobile phase was adjusted from 5% to 100% (*v*/*v*) in this study. Premature elution of DON from the C_18_ column and obvious overlap of chromatographic peaks between DON and endogenous interferences were observed when performing isocratic elution with a mobile phase containing a high content (>20%, *v*/*v*) of methanol or acetonitrile. On the other hand, significant reductions in analytical sensitivity and incomplete elution of some non-polar interferences were observed when the content of methanol or acetonitrile in the mobile phase was reduced to less than 8% (*v*/*v*). The incomplete elution of non-polar interferences might shorten the service life of the chromatographic column and interfere with the analysis of the next sample. No significant difference in chromatographic separation was observed when comparing acetonitrile–water mixtures with methanol–water and methanol–acetonitrile–water mixtures. Satisfactory chromatographic separation and analytical sensitivity and acceptable runtime were achieved when a gradient elution procedure (0–12.5 min, 8% acetonitrile, 12.5–16 min, 100% acetonitrile, and 16–20 min, 8% acetonitrile) was applied in our UPLC method (Figure 1).

The criterion for evaluating the detoxification efficacy of mycotoxin adsorbent is as follows: the higher the concentration and frequency of DON detected in pig feces, the superior the detoxification efficacy of mycotoxin adsorbent. This criterion is based on the toxicokinetic profiles of DON in pigs. After oral ingestion, DON is well absorbed in the pig intestine lumen. More than 55% of ingested DON can enter the blood and subsequently be transported to the target organs, thereby eliciting toxicological effects [7]. If the mycotoxin adsorbent can adsorb DON and form a DON–adsorbent complex that is difficult to be absorbed, the ingested DON will be excreted from the body along with feces, while effectively reducing the toxic effects of DON on pigs. The rationality of above criterion was also demonstrated in our animal experiment. As detailed in Table 3, nondetectable or low concentrations of DON (<1.38 μg/g) were found in fecal samples from groups A and B (negative control), while six out of twenty fecal batches from group C (adsorbent 1-treated) contained high concentrations of DON (>20 μg/g). This indicated that the effective adsorption of DON by adsorbent 1 occurred in the pig intestine lumen, thereby preventing the absorption of DON into the blood and effectively protecting the pigs. The concentration and frequency of DON detected in fecal samples from groups D (adsorbent 2-treated) and E (adsorbent 3-treated) were substantially lower than those of group C. This indicated that adsorbents 2 and 3 were unable to efficiently adsorb DON in the pig intestine lumen, thereby highlighting their poor detoxification efficacies. The reliability of our in vivo evaluation criterion was further validated in the subsequent adsorbent application trials at six pig farms. Basically, consistent results were observed in both the adsorbent application trials and our animal experiments. It should be noted that high concentrations of DON were still not detected in fourteen out of twenty fecal batches collected from the pigs in group C, even though these animals were treated with adsorbent 1. Similarly, no medium or high concentrations of DON were found in the fecal samples collected from pig farm 2, where adsorbent 1 was used. The reasons for these results included: (i) the uneven distribution of DON in mildewed feedstuffs made it difficult to ensure that an equal amount of DON was ingested during each feeding; (ii) the uneven distribution of mycotoxin adsorbent in mildewed feedstuffs resulted in an inconsistent adsorption efficacy; and (iii) the uneven distribution of DON in pig feces, coupled with the relatively small sample size (only 2 g per sample), made it challenging to prevent false negative results. Nevertheless, in a long-team animal experiment (such as 10 d), the quantity and frequency of DON detected in fecal samples from the pigs treated with an effective adsorbent were generally expected to be significantly higher than those in fecal samples from pigs either untreated with any adsorbent or treated with an ineffective one. In other words, the long-term use of the adsorbent to be evaluated and the collection of fecal samples in multiple batches are indispensable to guarantee the reliability of the evaluation results.

So far, there have been only a few studies that have described in vivo methods for evaluating the detoxification efficacy of mycotoxin adsorbents [23,26,27]. These methods use non-specific parameters or toxicokinetic profiles of mycotoxins and their metabolites as evaluation indicators. However, although useful, non-specific parameters are not sufficient to prove the detoxification efficacy [28], and obtaining the toxicokinetic data (plasma or tissue concentration–time data of DON) may be difficult and time consuming. In this study, a new strategy was proposed to evaluate the detoxifying efficacy of mycotoxin adsorbent in vivo by analyzing DON concentration in feces. From our perspective, the concentration and frequency of DON detected in pig feces serve as more direct and reliable indicators for evaluating the detoxification efficacy compared with non-specific parameters. On the other hand, the DON detected in feces represents the unabsorbed fraction, whereas the DON detected in blood or tissues indicates the absorbed fraction. The concentration and frequency of DON detected in feces appear to more directly reflect the detoxification efficacy of the adsorbent. Notably, fecal sample collection is a non-invasive procedure for pigs, which is easier than obtaining blood and tissue samples.

## 4. Conclusions

In summary, a simple and reliable UPLC method was developed and validated for the quantification of DON in feces and feedstuffs. A criterion for evaluating the detoxification efficacy of the mycotoxin adsorbent was proposed based on the toxicokinetics of DON in pigs and our animal experimental results. That is, the higher the concentration and frequency of DON detected in pig feces, the superior the detoxification efficacy of the mycotoxin adsorbent. We believe our findings may provide a possible strategy to evaluate the detoxifying efficacy of mycotoxin adsorbents in vivo.

## 5. Materials and Methods

### 5.1. Chemicals and Reagents

Analytical standard of DON (powder form, purity > 95.0%) was purchased from Toronto Research Chemicals Co., Ltd. (Toronto, ON, Canada). LC/MS-grade acetonitrile was supplied by Merck KGaA (Darmstadt, Germany), and analytical grade acetonitrile, methanol, ethyl acetate, ethanol, acetone, 70% methanol solution (*v*/*v*), 84% acetonitrile solution (*v*/*v*), isopropyl alcohol, dichloromethane, and dimethyl sulfoxide were sourced from Sinopharm Chemical Reagent (Shanghai, China). Ultrapure water with a resistivity of 18.3 MΩ×cm was used throughout the experiments. A stock solution of DON, at 2500 μg/mL, was prepared in acetonitrile and stored at −20 °C. Working standards were prepared every week by diluting the stock solution in 10% acetonitrile solution (*v*/*v*) to obtain desired concentrations. These working standards were also stored at −20 °C.

### 5.2. Instruments and Materials

A Dionex Ultimate 3000 UPLC System (Thermo Fischer Scientific, Waltham, MA, USA) was used for the chromatographic determination of DON in fecal and feedstuff samples. The nylon syringe filter (0.45 μm, 13 mm) used for sample purification was purchased from Lizhu Biological Technology Co., Ltd. (Guangzhou, China).

Pig feces were obtained from two sources: a commercial pig farm (Putian Youlike Agriculture and Animal Husbandry Development Co., Ltd. (Putian, China)) and our animal experiment. Fecal samples from the commercial pig farm were confirmed to be negative for DON and then used for the validation of the analytical method. Fecal samples from our animal experiments (incurred samples) were used to evaluate the detoxification efficacies of commercial adsorbents.

The pig feedstuffs (commercial feedstuffs) were obtained from Putian Youlike Agriculture and Animal Husbandry Development Co., Ltd. (Putian, China). These feedstuffs were confirmed to be negative for DON too. The DON-contaminated (mildewed) feedstuffs were prepared by placing the commercial feedstuffs in an environment with suitable temperature (20–25 °C) and humidity (70–100%) for 9 d. These feedstuffs were turned and mixed daily to make DON distribute as evenly as possible in them. Aliquots (5 g) of such feedstuff samples were taken at 1, 3, 5, and 9 d, respectively, and analyzed by UPLC to determine the concentrations of DON.

The mycotoxin adsorbents used in this study were purchased from the market. For ease of distinction, these mycotoxin adsorbents were named adsorbent 1 (Alquerfeed Antitox^®^, light brown powder, active ingredient: zeolite powder and silicoglycidol), adsorbent 2 (JialibaiLing^®^, light gray to brown/black, free-flowing powder, active ingredient: calcium montmorillonite), and adsorbent 3 (MeikeXi^®^, light tan free-flowing powder, active ingredient: brewer’s dried yeast, calcium carbonate, yeast cell walls, and hydrated sodium calcium aluminosilicate), respectively.

### 5.3. Animals

Fifteen healthy crossbred (duroc × landrace × large white) pigs, weighing 26.3 ± 3.9 kg, were purchased from Putian Youlike Agriculture and Animal Husbandry Development Co., Ltd. (Putian, China). The pigs were clinically healthy based on physical examination, hematologic evaluation, and urinalysis. Each pig was housed separately in a metabolic cage (1.2 m × 0.5 m × 0.4 m). These cages were placed in two well-ventilated rooms with suitable temperature (24.8 ± 2.9 °C) and humidity (70.2–86.3%). A 7 d acclimatization period was provided before the experiment. During the acclimatization period, the pigs were allowed ad libitum access to water and feedstuffs without DON. The animal experimental design and protocol used in this study were approved by the Research Ethics Committee of the College of Animal Science, Fujian Agriculture and Forestry University (No. PZCASFAFU18046).

### 5.4. In Vivo Evaluation of the Detoxification Efficacies of Three Commercial Mycotoxin Adsorbents

#### 5.4.1. Analytical Method

Fresh feces were dried in an oven at 60 °C for 24 h and then finely ground to pass through a 50-mesh sieve. Dried feces powder (150 ± 1.5 mg) was weighed into a 1.5 mL polypropylene centrifuge tube, and DON was added at three quality control (QC) concentrations (0.1, 1, and 10 μg/g). The analyte was omitted from the incurred samples. Next, the samples were processed according to a procedure modified from Miró-Abella et al. [37,38]. In brief, 0.9 mL of acetonitrile was added. The mixture was vortexed for 3 min, followed by ultrasonic extraction at 60 °C for 10 min. After centrifugation at 12,000 rpm for 2 min, the supernatant was transferred to a new 1.5 mL centrifuge tube. Another 0.6 mL of acetonitrile was added to the residue, and the extraction procedure was repeated once more. The supernatants were combined and evaporated to dryness at 60 °C, under nitrogen flow. The resultant residue was reconstituted in 0.3 mL of 30% acetonitrile solution (*v*/*v*) and filtered through a 0.45 μm nylon syringe filter for subsequent UPLC analysis. The pig feedstuffs were ground to pass through a 50-mesh sieve. The feedstuff powder (300 ± 3 mg) was weighed in a 1.5 mL polypropylene centrifuge tube, and DON was added at three QC concentrations (0.01, 0.1, and 1 μg/g). The analyte was omitted from the incurred samples. Next, the sample was processed as described above.

The analytical conditions were as follows: chromatographic column, Agilent Zorbax SB C_18_ column (4.6 × 250 mm, 5 μm) (Agilent Technologies, Santa Clara, CA, USA); column temperature, 35 °C; mobile phase A, acetonitrile; mobile phase B, water; chromatographic elution mode, gradient (0–12.5 min, 8% A, 12.5–16 min, 100% A, and 16–20 min, 8% A); flow rate, 1 mL/min; injection volume, 20 μL; ultraviolet detector, 220 nm.

The analytical performance of the UPLC method was evaluated in terms of specificity, linearity, sensitivity, accuracy, precision, and stability [39]. The specificity was validated by comparing the chromatograms of blank and spiked fecal and feedstuff samples. The concentrations of DON in fecal and feedstuff samples were quantified using a standard curve, which was generated by plotting the peak area of DON against the corresponding concentration. The linearity of the standard curve was evaluated by the correlation coefficient (r). The sensitivity was represented by the limit of detection (LOD) and the limit of quantification (LOQ). Twenty blank samples were processed as described above. The resulting blank sample extracts were spiked with DON standards at eight concentrations from 0.02 to 50 μg/mL. Then, these spiked samples were analyzed as described above. The LOD and LOQ were the lowest concentrations with a signal-to-noise ratio of 3 and 10, respectively. The precision of the method was represented by the intra-day and inter-day relative standard deviation (RSD), and the analytical accuracy was represented by the mean recovery. Five consecutive analytical batches were performed, each including three concentrations of QC samples with five replicates for each concentration, as described above to calculate these parameters. The storage stabilities of DON in fecal and feedstuff samples were also evaluated under various conditions (7 d at −20 °C and 2 d at 60 °C).

#### 5.4.2. Experimental Design

Fifteen pigs were randomly assigned to five groups with three replicates per group, namely group A, group B, group C, group D, and group E. The pigs were fed twice a day, for 10 days with 500 g of the following: commercial feedstuffs (group A); mildewed feedstuffs (group B); mildewed feedstuffs containing 0.2% (*w*/*w*) adsorbent 1 (group C); mildewed feedstuffs containing 0.2% (*w*/*w*) adsorbent 2 (group D); and mildewed feedstuffs containing 0.2% (*w*/*w*) adsorbent 3 (group E). For each pig, a 2 g fecal sample was collected before each feeding (during the adsorbent administration period, fecal samples were collected daily at 8 a.m. and 6 p.m.). These samples were placed separately in marked containers and stored at −20 °C until UPLC analysis. The remaining feces were removed to prevent interference with the next sample. During the experiment period, the concentrations of DON in all feedstuff samples were also determined using our UPLC method, and the body weight and clinical manifestation of each pig were recorded.

To further validate the reliability of the in vivo evaluation criterion, adsorbents 1 and 2 were applied to six pig farms located in Fujian Province, China. Specifically, adsorbent 1 was used in pig farms 1–3, while adsorbent 2 was applied in pig farms 4–6. The feed samples from six pig farms (five replicates for each) were analyzed by our UPLC method to determine the concentration of DON before the experiment. During the two-week period of continuous adsorbent application, a total of 31 fecal samples were collected from these pig farms. The concentrations of DON in these samples were analyzed by our UPLC method. Subsequently, the detoxification effect of the two adsorbents was evaluated based on the concentration and frequency of DON detected in these fecal samples.

#### 5.4.3. Data Analysis

The concentrations of DON in fecal and feedstuff samples, recoveries, RSD, and all other data were expressed as the means ± standard deviation (SD). The differences in body weight among five groups were determined using a 2-tailed *t*-test (* *p* < 0.05, ** *p* < 0.01). The statistical analyses were performed using SPSS version 21 (IBM Co., Armonk, NY, USA).

## Figures and Tables

**Figure 1 toxins-17-00322-f001:**
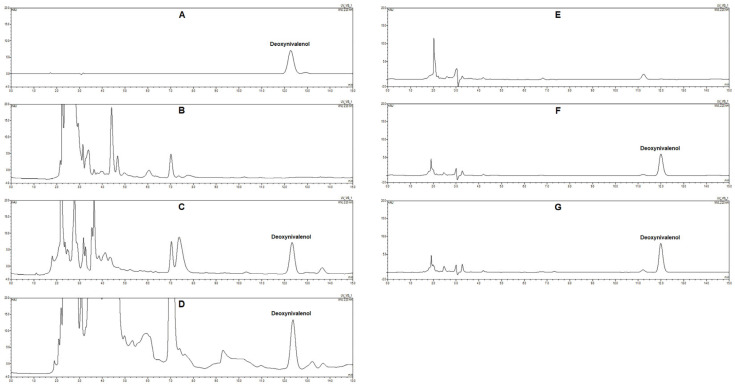
Representative chromatograms of deoxynivalenol in pig feces and pig feedstuffs. (**A**) deoxynivalenol standard (1 μg/mL); (**B**) blank feedstuff sample; (**C**) blank feedstuff sample spiked with deoxynivalenol at 1 μg/g; (**D**) incurred feedstuff sample collected on day 9 during the preparation of mildewed feedstuffs; (**E**) blank fecal sample; (**F**) blank fecal sample spiked with deoxynivalenol at 1 μg/g; (**G**) incurred fecal sample collected from one pig in group E on day 1 during the animal experiment.

**Figure 2 toxins-17-00322-f002:**
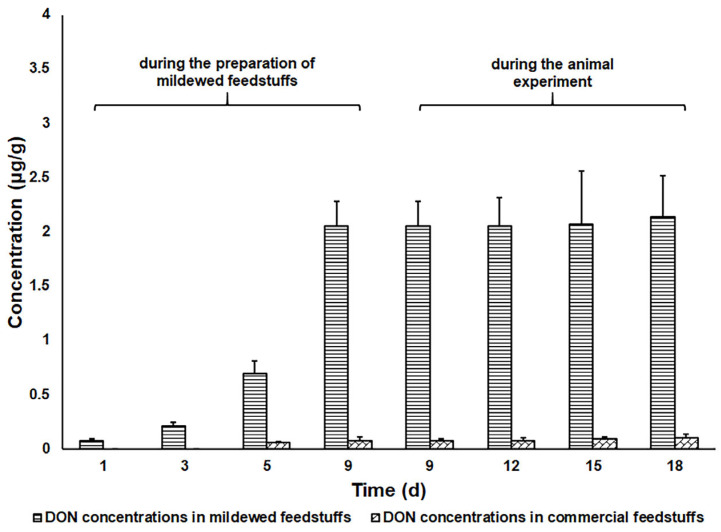
The concentrations of DON in commercial and mildewed feedstuffs, during the preparation of mildewed feedstuffs and the animal experiment.

**Table 1 toxins-17-00322-t001:** Precision and accuracy of the UPLC method for determining of DON in pig feces and pig feedstuffs.

Matrix	Spiked Concentration (μg/g)	Recovery (%) ^a^	Intra-Day RSD (%)	Inter-Day RSD (%)
Pig feces	0.1	86.4 ± 0.4	0.4	0.8
		87.0 ± 0.5	0.6	
		86.8 ± 0.4	0.5	
		86.7 ± 0.5	0.6	
		86.3 ± 1.1	1.3	
	1	98.8 ± 1.4	1.4	1.2
		97.5 ± 0.3	0.3	
		97.7 ± 1.3	1.3	
		97.2 ± 0.4	0.5	
		98.1 ± 1.1	1.1	
	10	104.1 ± 1.3	1.2	0.3
		105.3 ± 0.4	0.4	
		104.5 ± 0.9	0.9	
		104.1 ± 1.7	1.7	
		105.1 ± 1.5	1.4	
Pig feedstuffs	0.01	105.3 ± 15.4	14.6	11.7
		108.6 ± 8.5	7.8	
		105.3 ± 12.6	12.0	
		105.5 ± 6.8	6.4	
		106.6 ± 15.0	14.0	
	0.1	93.9 ± 6.1	6.5	5.6
		94.6 ± 6.0	6.3	
		96.3 ± 1.4	1.5	
		92.6 ± 3.8	4.1	
		91.8 ± 5.4	5.9	
	1	97.7 ± 1.7	1.7	2.0
		97.3 ± 2.6	2.7	
		97.3 ± 2.4	2.5	
		96.9 ± 0.8	0.8	
		96.9 ± 1.6	1.6	

^a^ Mean ± SD, n = 5.

**Table 2 toxins-17-00322-t002:** Storage stability of DON in pig feces and pig feedstuffs.

Matrix	Spiked Concentration (μg/g)	Condition	Recovery (%) ^a^	RSD (%)
Pig feces	0.1	−20 °C, 7 d	92.4 ± 5.2	5.6
		60 °C, 2 d	93.8 ± 8.4	9.0
	1	−20 °C, 7 d	95.6 ± 7.6	7.9
		60 °C, 2 d	98.7 ± 4.4	4.5
	10	−20 °C, 7 d	100.3 ± 5.1	5.1
		60 °C, 2 d	96.9 ± 5.5	5.7
Pig feedstuffs	0.01	−20 °C, 7 d	100.2 ± 3.7	3.7
		60 °C, 2 d	95.8 ± 7.4	7.7
	0.1	−20 °C, 7 d	91.6 ± 5.2	5.7
		60 °C, 2 d	90.7 ± 5.8	6.4
	1	−20 °C, 7 d	94.3 ± 2.5	2.7
		60 °C, 2 d	88.9 ± 4.0	4.6

^a^ Mean ± SD, n = 5.

**Table 3 toxins-17-00322-t003:** The concentrations of DON in pig feces (μg/g) throughout the animal experiment.

Time (d)	Sampling Time	Group A ^a^	Group B ^a^	Group C ^a^	Group D ^a^	Group E ^a^
1	8 a.m.	ND	0.06 ± 0.13	8.68 ± 2.56	5.54 ± 3.21	ND
	6 p.m.	ND	0.06 ± 0.16	0.5 ± 1.16	ND	1.52 ± 1.01
2	8 a.m.	0.06 ± 0.12	0.19 ± 1.3	0.28 ± 0.14	0.60 ± 1.13	0.52 ± 0.42
	6 p.m.	ND	ND	2.26 ± 1.77	ND	ND
3	8 a.m.	ND	1.00 ± 1.23	ND	0.36 ± 0.19	ND
	6 p.m.	ND	0.10± 0.12	ND	ND	ND
4	8 a.m.	ND	0.26 ± 0.14	0.4 ± 1.08	0.14 ± 0.15	0.38 ± 0.16
	6 p.m.	ND	ND	0.52 ± 0.19	ND	ND
5	8 a.m.	0.14 ± 0.12	0.26 ± 0.16	66.28 ± 68.34	ND	ND
	6 p.m.	0.38 ± 0.20	0.14 ± 0.12	71.06 ± 67.77	ND	0.48 ± 0.26
6	8 a.m.	0.9 ± 0.14	0.62 ± 0.18	0.34 ± 0.12	ND	6.5 ± 5.73
	6 p.m.	1.1 ± 0.16	ND	91.92 ± 89.95	1.92 ± 1.02	6.38 ± 4.35
7	8 a.m.	ND	0.32 ± 0.40	0.72 ± 0.18	ND	ND
	6 p.m.	0.06 ± 0.11	0.04 ± 0.11	23.22 ± 18.45	0.66 ± 1.21	1.3 ± 1.52
8	8 a.m.	1.38 ± 0.16	ND	0.7 ± 0.16	0.14 ± 0.12	0.86 ± 1.22
	6 p.m.	1.3 ± 1.8	0.26 ± 0.11	26.24 ± 26.75	0.82 ± 1.24	0.04 ± 0.11
9	8 a.m.	0.2 ± 0.11	0.68 ± 1.9	30.22 ± 27.38	ND	0.20 ±0.16
	6 p.m.	0.34 ± 1.1	0.46 ± 1.7	ND	0.26 ± 0.13	0.48 ± 0.11
10	8 a.m.	ND	0.19 ± 0.12	0.22 ± 0.12	ND	ND
	6 p.m.	ND	0.22 ± 0.11	0.12 ± 0.12	0.16 ± 0.13	ND

^a^ Mean ± SD, n = 5. ND: not detected.

**Table 4 toxins-17-00322-t004:** The concentrations of DON in fecal samples (μg/g) from six pig farms.

Sample No.	Pig Farm 1	Pig Farm 2	Pig Farm 3	Pig Farm 4	Pig Farm 5	Pig Farm 6
1	ND	0.05	0.07	0.17	0.08	0.04
2	0.03	0.07	0.05	0.03	0.13	0.22
3	27.72	0.38	20.17	0.37	0.31	1.44
4	0.09	1.87	12.10	0.21	0.44	0.07
5	0.09	0.83	0.46	0.44	0.74	0.47
6	-	-	7.96	-	-	-

ND: not detected.

## Data Availability

The original contributions presented in this study are included in the article. Further inquiries can be directed to the corresponding authors.

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
