# Peer review of "Validation of a Methodology for the Quantification of DON in Feces and Feedstuffs by UPLC as Possible Strategy to Evaluate the Detoxifying Efficacy of a Mycotoxin Adsorbent In Vivo"

_toxins, 2025, doi:10.3390/toxins17070322_

Round 1

Reviewer 1 Report

Comments and Suggestions for Authors

This is an interesting paper with a suitable length and organization.

However, I have doubts about the general approach of this paper mainly with regards to the title, objective and the conclusions obtained. As stated in these sections, a method has been developed and validated to evaluate in vivo the detoxification capacity of mycotoxin adsorbents in feed. However, this is not exactly what it has been demonstrated. What has been validated is a methodology to quantify DON in faecal samples by UPLC. Although the results obtained are of interest, they are not enough to say that these analysis are a reliable simple and effective method for evaluating the detoxification efficacies of sorbents. To be able to say this, it would be necessary to carry out more studies to know exactly whether the non-detection of DON in faeces is because the adsorbent did not work or for what other reasons. What is more, in the case in which adsorbent is supposed to have worked (group C) results greatly differ even in the same day of analysis.

I believe that this article contains results that could be the beginning of a line of research that attempts to standardize faecal analysis as a measure of the efficacy of adsorbents for mycotoxins in animal feed. However, it cannot be concluded from what is presented here that these analysis can be used to demonstrate the detoxification capacity of adsorbents in a simple and reliable way.

Tittle

I don't find the title reflects the objective of the paper. A new methodology for the in vivo evaluation of the detoxifying efficacy of a mycotoxin adsorbent has not been developed. An UPLC methodology for analyzing DON in faeces have just been developed and validated. Therefore, the tittle should be changed as for example: Validation of a methodology for the quantification of DON in faeces by UPLC as possible strategy to in vivo evaluate the detoxifying efficacy of a mycotoxin adsorbent.

Introduction section

Since Fusarium toxins are mostly formed in the preharvest stage, I think this sentence (lines 39-40) should be completed: the contamination of feedstuffs with DON is difficult to avoid, even under good agricultural practices, storage and transportation conditions.

I don’t agree with the sentence (lines 46-47): “An important reason for this is the absence of scientific evaluation method for the detoxification efficacies of these adsorbents”. There is not an accredited methodology for evaluating the detoxification efficacies but yes methods that are scientifically based. The same has been written in line 139 in the discussion section.

It is said (lines 65-67) that “When DON and mycotoxin adsorbents are orally ingested by pigs, the concentrations of DON in pig feces can better reflect the adsorption capacity (detoxification efficacy) of the adsorbents in vivo.” On what basis do the authors make this assertion?

Results

Table 1. It is said in the results and material section that for validating methodology have been analyzed: “Three consecutive analytical batches, including three concentrations of  QC samples (five replicates for each concentration)”. As regards, I guess that what is depicted in the column “Recovery” of Table 1 represent the results of each of the five replicates. I therefore do not understand where the value of the standard deviation of each of these 5 replicates comes from.

When using adsorbents, a DON-adsorbent complex will be excreted from the body along with feces. How the authors know if they have been capable of separating the DON of the adsorbent during the chromatographic analysis of the faeces?. It could be also the reason to not detecting DON in faeces. I think that it should be discussed. Perhaps the methodogy should have been validated with DON mycotoxin added to blank samples joined to adsorbents.

I don’ think that results corresponding to the “Validation of the in vivo evaluation method” should be included. As the feed consumed is not reported to contain DON it is not possible to assess the efficacy or otherwise of the adsorbents administered with the results that have been presented.

What exactly are the advantages of this method compared to other procedures for in vivo assessment of detoxification capacity should be discuss in a more in-depth way.

Conclusions

The methodology proposed allows the comparison of one adsorbent versus another but does not allow quantification of the efficiency of the adsorbent under evaluation. Therefore I don’t agree with the conclusions stated.

Material and methods

I understand that it is an error the text in lines 231 to 251 and that it is going to be deleted.

5.4.2. Experimental design. I don’t think authors must repeat so many times “were fed twice a day with 500 g”. I think you should better write:  The pigs were fed twice a day, for 10 days with 500 g of the following: commercial feedstuffs (group A); mildewed feedstuffs (group B); mildewed feedstuffs containing 0.2% (w/w) adsorbent 1 and so on…

In addition to the fecal samples collected prior to the adsorbent administration trials, the authors should specify on which days and at which times fecal samples have been collected to evaluate the effect of the adsorbents.

When the experimental design is described it should be explained in detail which mycotoxigenic mold strain has been used for the preparation of mildewed feedstuffs.

Author Response

Responses to Reviewer 1:

  1. This is an interesting paper with a suitable length and organization. However, I have doubts about the general approach of this paper mainly with regards to the title, objective and the conclusions obtained. As stated in these sections, a method has been developed and validated to evaluate in vivo the detoxification capacity of mycotoxin adsorbents in feed. However, this is not exactly what it has been demonstrated. What has been validated is a methodology to quantify DON in faecal samples by UPLC. Although the results obtained are of interest, they are not enough to say that these analysis are a reliable simple and effective method for evaluating the detoxification efficacies of sorbents. To be able to say this, it would be necessary to carry out more studies to know exactly whether the non-detection of DON in faeces is because the adsorbent did not work or for what other reasons. What is more, in the case in which adsorbent is supposed to have worked (group C) results greatly differ even in the same day of analysis. I believe that this article contains results that could be the beginning of a line of research that attempts to standardize faecal analysis as a measure of the efficacy of adsorbents for mycotoxins in animal feed. However, it cannot be concluded from what is presented here that these analysis can be used to demonstrate the detoxification capacity of adsorbents in a simple and reliable way.

Response: Thank you very much for your constructive comments. In this study, a simple and reliable UPLC method was developed and applied to detect DON in pig feces. Based on the toxicokinetics of DON in pigs and our animal experiment results, we proposed a new strategy to evaluate the detoxifying efficacy of mycotoxin adsorbents in vivo. Specifically, the detoxifying efficacy of adsorbent was determined by DON detection in pig feces. Nevertheless, we agree with your doubts regarding the title, objective and conclusions drawn in this study. The descriptions in these sections of the original manuscript could be improved and would benefit from more conservative phrasing. We have revised the corresponding sections in the revised manuscript (please refer to lines 2–4, lines 16–17, lines 33–36, lines 86–88, lines 156–161, line 162, line 236, lines 270–276, and line 380). We hope such changes can make the description more appropriate.

  1. I don't find the title reflects the objective of the paper. A new methodology for the in vivo evaluation of the detoxifying efficacy of a mycotoxin adsorbent has not been developed. An UPLC methodology for analyzing DON in faeces have just been developed and validated. Therefore, the tittle should be changed as for example: Validation of a methodology for the quantification of DON in faeces by UPLC as possible strategy to in vivo evaluate the detoxifying efficacy of a mycotoxin adsorbent.

Response: We have revised the corresponding sections in the revised manuscript (please refer to lines 2–4). We hope this meets your requirements.

  1. Since Fusarium toxins are mostly formed in the preharvest stage, I think this sentence (lines 39-40) should be completed: the contamination of feedstuffs with DON is difficult to avoid, even under good agricultural practices, storage and transportation conditions.

Response: We have revised the corresponding sections in the revised manuscript (please refer to lines 49–51). We hope this meets your requirements.

  1. I don’t agree with the sentence (lines 46-47): “An important reason for this is the absence of scientific evaluation method for the detoxification efficacies of these adsorbents”. There is not an accredited methodology for evaluating the detoxification efficacies but yes methods that are scientifically based. The same has been written in line 139 in the discussion section.

Response: We have revised the corresponding sections in the revised manuscript (please refer to lines 59–61 and lines 156–158). We hope such change can make the description more appropriate.

  1. It is said (lines 65-67) that “When DON and mycotoxin adsorbents are orally ingested by pigs, the concentrations of DON in pig feces can better reflect the adsorption capacity (detoxification efficacy) of the adsorbents in vivo.” On what basis do the authors make this assertion?

Response: DON is well absorbed in the pig intestine with a bioavailability exceeding 55% [1]. However, if DON in feedstuff is successfully adsorbed by the adsorbents, the bound DON (DON-adsorbent complex) will be difficult to be absorbed into the bloodstream, thereby remaining in the intestine and eventually being excreted with feces [2]. Therefore, the amount and frequency of DON detected in feces can reflect the in vivo detoxification efficacy of adsorbent. That is, higher DON concentration and frequency detected in feces indicate better adsorbent efficacy.

References

[1] Sun Y, Jiang J, Mu P, Lin R, Wen J, Deng Y. Toxicokinetics and metabolism of deoxynivalenol in animals and humans. Arch Toxicol. 2022 Oct;96(10):2639-2654. doi: 10.1007/s00204-022-03337-8.

[2] Holanda DM, Kim SW. Mycotoxin Occurrence, Toxicity, and Detoxifying Agents in Pig Production with an Emphasis on Deoxynivalenol. Toxins (Basel). 2021 Feb 23;13(2):171. doi: 10.3390/toxins13020171.

  1. Table 1. It is said in the results and material section that for validating methodology have been analyzed: “Three consecutive analytical batches, including three concentrations of QC samples (five replicates for each concentration)”. As regards, I guess that what is depicted in the column “Recovery” of Table 1 represent the results of each of the five replicates. I therefore do not understand where the value of the standard deviation of each of these 5 replicates comes from.

Response: There was a mistake in the description of the analytical performance of the UPLC method in Section 5.4.1 of the original manuscript. The correct description is: Five consecutive analytical batches, including three concentrations of QC samples (with five replicates per concentration), were conducted as described above to calculate these parameters. We do apologize for mistake like this. We have revised the corresponding sections in the revised manuscript (please refer to lines 363–365).

  1. When using adsorbents, a DON-adsorbent complex will be excreted from the body along with feces. How the authors know if they have been capable of separating the DON of the adsorbent during the chromatographic analysis of the faeces?. It could be also the reason to not detecting DON in faeces. I think that it should be discussed. Perhaps the methodogy should have been validated with DON mycotoxin added to blank samples joined to adsorbents.

Response: During the analysis of fecal samples, if the extraction solvent fails to efficiently separate DON from the DON-adsorbent complex, it can indeed lead to false negative results, thereby causing misjudgment of the detoxifying efficacy of mycotoxin adsorbent (underestimating its detoxification efficacy). Although most adsorbents bind DON through weak intermolecular forces [1], further evaluation of these solvents’ efficiencies for separating DON from DON-adsorbent complexes remains necessary. Based on these considerations, we conducted an experiment to evaluate the efficiency of acetonitrile for separating DON from the DON-adsorbent complex. The results showed that acetonitrile effectively separated DON from the complex, yielding a recovery rate of 98.3 ± 3.2%. Accordingly, we have added relevant content to the Discussion section of the revised manuscript (please refer to lines 186–198). We hope this meets your requirements.

References

[1] Cavret S, Laurent N, Videmann B, Mazallon M, Lecoeur S. Assessment of deoxynivalenol (DON) adsorbents and characterisation of their efficacy using complementary in vitro tests. Food Addit Contam Part A Chem Anal Control Expo Risk Assess. 2010 Jan;27(1):43-53. doi: 10.1080/02652030903013252.

  1. I don’ think that results corresponding to the “Validation of the in vivo evaluation method” should be included. As the feed consumed is not reported to contain DON it is not possible to assess the efficacy or otherwise of the adsorbents administered with the results that have been presented.

Response: The feed samples from six pig farms (five replicates for each) were analysed by our UPLC method to determine the concentration of DON before the experiment. The results showed that the DON concentrations in feed samples from six pig farms were 1.55 ± 0.32 μg/g (farm 1), 2.74 ± 0.58 μg/g (farm 2), 1.89 ± 0.49 μg/g (farm 3), 0.87 ± 0.26 μg/g (farm 4), 3.24 ± 0.27 μg/g (farm 5), and 1.21 ± 0.33 μg/g (farm 6). We have supplemented these experimental data in the revised manuscript and revised the corresponding sections (please refer to lines 125–127 and lines 382–384).

  1. What exactly are the advantages of this method compared to other procedures for in vivo assessment of detoxification capacity should be discuss in a more in-depth way.

Response: So far, only a few studies [1–3] have described in vivo methods for evaluating the detoxification efficacies of mycotoxin adsorbents. These methods evaluate the detoxification efficacies based on non-specific parameters such as growth performance, hepatic function, and inflammatory status [2], or the toxicokinetic profiles of mycotoxins and their metabolites in the body [1, 3]. However, the non-specific parameters, although useful, are not sufficient to prove the detoxification efficacies [4], and obtaining the toxicokinetic data (plasma or tissue concentration-time data of DON) may be difficult and time-consuming. In this study, a new strategy was proposed to evaluate the detoxifying efficacy of mycotoxin adsorbents in vivo by analyzing DON concentration in feces. From our perspective, the concentration and frequency of DON detected in pig feces serve as more direct and reliable indicators for evaluating the detoxification efficacy compared with nonspecific parameters. On the other hand, the DON detected in feces represents the unabsorbed fraction, whereas the DON detected in blood or tissues indicates the absorbed fraction. The concentration and frequency of DON detected in feces appear to more directly reflect the detoxification efficacy of adsorbent. Notably, fecal sample collection is a non-invasive procedure for pigs, which is easier than obtaining blood and tissue samples. We have added these contents to the Discussion section of the revised manuscript (please refer to lines 254–268). We hope this will better highlight the value of this study.

References

[1] Bruinenberg PG, Castex M. Evaluation of a Yeast Hydrolysate from a Novel Strain of Saccharomyces cerevisiae for Mycotoxin Mitigation using In Vitro and In Vivo Models. Toxins (Basel). 2021 Dec 22;14(1):7. doi: 10.3390/toxins14010007.

[2] Holanda DM, Kim YI, Parnsen W, Kim SW. Phytobiotics with Adsorbent to Mitigate Toxicity of Multiple Mycotoxins on Health and Growth of Pigs. Toxins (Basel). 2021 Jun 26;13(7):442. doi: 10.3390/toxins13070442.

[3] Lauwers M, Croubels S, Letor B, Gougoulias C, Devreese M. Biomarkers for Exposure as A Tool for Efficacy Testing of A Mycotoxin Detoxifier in Broiler Chickens and Pigs. Toxins (Basel). 2019 Mar 28;11(4):187. doi: 10.3390/toxins11040187.

[4] EFSA Panel on Additives and Products or Substances used in Animal Feed (FEEDAP). EFSA Statement on the establishment of guidelines for the assessment of additives from the functional group ‘substances for reduction of the contamination of feed by mycotoxins’. EFSA J. 2010, 8, 1963.

  1. The methodology proposed allows the comparison of one adsorbent versus another but does not allow quantification of the efficiency of the adsorbent under evaluation. Therefore I don’t agree with the conclusions stated.

Response: We agree with your comments regarding the conclusion. We have revised the Conclusions section in the revised manuscript (please refer to lines 270–276). We hope such change can make the description more appropriate.

  1. I understand that it is an error the text in lines 231 to 251 and that it is going to be deleted.

Response: We have deleted the corresponding section.

  1. 4.2. Experimental design. I don’t think authors must repeat so many times “were fed twice a day with 500 g”. I think you should better write:  The pigs were fed twice a day, for 10 days with 500 g of the following: commercial feedstuffs (group A); mildewed feedstuffs (group B); mildewed feedstuffs containing 0.2% (w/w) adsorbent 1 and so on…

Response: We have revised the corresponding section according to you (please refer to lines 369–373).

  1. In addition to the fecal samples collected prior to the adsorbent administration trials, the authors should specify on which days and at which times fecal samples have been collected to evaluate the effect of the adsorbents.

Response: The sampling time is listed in Table 3 (please refer to lines 150–151). We have added a sentence to specify the detailed sampling time (please refer to lines 373–375). We hope such a change can make the description clearer.

  1. When the experimental design is described it should be explained in detail which mycotoxigenic mold strain has been used for the preparation of mildewed feedstuffs.

Response: To simulate the real-world scenario of mold contamination in feedstuff, we did not deliberately inoculate the commercial feedstuffs with mycotoxin-producing fungal strains or spores. Instead, the feedstuffs were exposed to natural environmental conditions (temperature (20–25 °C), humidity (70–100%)) to allow spontaneous colonization by indigenous molds, thereby leading to natural mildewing. We hope this explanation meets your requirements.

Reviewer 2 Report

Comments and Suggestions for Authors

The manuscript submitted to me for review concerns the development of a new method for the detoxification of animal feed contaminated with mycotoxin deoxynivalenol (DON) by adding appropriate adsorbents and using the UPLC method to determine the content of DON in animal feces and feed samples. The article is interesting and addresses the important issue of eliminating mycotoxins from feed and thus protecting animals from the negative effects of mycotoxicosis and economic losses for farmers. However, the manuscript requires corrections and should be improved.

The introduction should include examples of adsorbents used so far (L48-50) and adsorption mechanisms. The last sentence of L73-74 should be rewritten to emphasize the purpose of the work.

The results are well described and sufficiently presented. In the discussion, the authors focused on the meticulous interpretation of their research results, but there are too few references to the literature. There are only a few citations regarding the use of chromatographic methods for mycotoxin analysis (L167-170) and the description of what happens to DON in the animal's body (L188-180). I am sure that it is worth comparing the results of this new method with those of other researchers, even if the methods are different, to indicate the advantages of presented method and the observed difficulties related to mycotoxin capture and prevention of its absorption.

In conclusions, the authors should base their own results that they obtained and in what specific ways their method can be helpful beyond the current state of knowledge and application.

In the Materials and methods section (L237-251) there is still an instruction for authors, which should be removed. L282-283 – There is a lack the trade names of the adsorbents and a brief characteristic (chemical nature). In subsection 5.4 there are no references to literature, which suggests that the authors independently developed the protocols for preparing stool and feed samples for extraction and independently designed the entire experiment. However, I think that there is a reference that they followed regarding methodological details and it should be included. All the above aspects must be taken into account by the authors for the manuscript to be complete and suitable for publication.

Author Response

We appreciate the reviewers’ valuable comments very much, which are helpful to improve the quality of our present study. We have revised our manuscript in accordance with these comments. To make it easier to read, we have highlighted the changes within the revised manuscript by using coloured (red) text.

Responses to Reviewer 2:

  1. The introduction should include examples of adsorbents used so far (L48-50) and adsorption mechanisms. The last sentence of L73-74 should be rewritten to emphasize the purpose of the work.

Response: Thank you very much for your positive and constructive comments. We have added relevant content in the Introduction section of the revised manuscript (please refer to lines 56–59). We have also rewritten the last sentence of the introduction according to you (please refer to lines 86–88). We hope this meets your requirements.

  1. The results are well described and sufficiently presented. In the discussion, the authors focused on the meticulous interpretation of their research results, but there are too few references to the literature. There are only a few citations regarding the use of chromatographic methods for mycotoxin analysis (L167-170) and the description of what happens to DON in the animal's body (L188-180). I am sure that it is worth comparing the results of this new method with those of other researchers, even if the methods are different, to indicate the advantages of presented method and the observed difficulties related to mycotoxin capture and prevention of its absorption.

Response: We have added a new paragraph to the Discussion section of the revised manuscript to compare our strategy with existing methods in terms of their advantages and disadvantages (please refer to lines 254–268). We hope this meets your requirements.

  1. In conclusions, the authors should base their own results that they obtained and in what specific ways their method can be helpful beyond the current state of knowledge and application.

Response: We have rewritten the Conclusions section in the revised manuscript (please refer to lines 270–276). We hope such change can make the description more appropriate.

  1. In the Materials and methods section (L237-251) there is still an instruction for authors, which should be removed.

Response: We have deleted the corresponding section.

  1. L282-283 – There is a lack the trade names of the adsorbents and a brief characteristic (chemical nature).

Response: We have supplemented information on the trade names, brief characteristic, and active ingredients of these adsorbents in Section 5.2 of the revised manuscript (please refer to lines 308–314).  We hope this meets your requirements.

  1. In subsection 5.4 there are no references to literature, which suggests that the authors independently developed the protocols for preparing stool and feed samples for extraction and independently designed the entire experiment. However, I think that there is a reference that they followed regarding methodological details and it should be included. All the above aspects must be taken into account by the authors for the manuscript to be complete and suitable for publication.

Response: We have supplemented two relevant references in Section 5.4 of the revised manuscript (please refer to lines 332–334).  We hope this meets your requirements.

Round 2

Reviewer 1 Report

Comments and Suggestions for Authors

The authors have made modifications in response to all the suggestions made.

However, in relation to the presentation of the results there is one issue on which I still have doubts and I would like the authors to review it again. It refers to Table 1. I don't understand what the results of the intraday recovery study correspond to. Do they mean that they tested 5 replicates of 5 samples fortified at 0.1 tested on different days; 5 replicates of 5 samples fortified at 1 on different days and 5 replicates of 5 samples fortified at 1 on different days? If so, why do you do it 5 times? and are the results of these 5 replicate 5-day samples used to calculate the interday recovery? On which organization or document did you base your approach to method validation?

Author Response

Responses to Reviewer 1:

  1. The authors have made modifications in response to all the suggestions made. However, in relation to the presentation of the results there is one issue on which I still have doubts and I would like the authors to review it again. It refers to Table 1. I don't understand what the results of the intraday recovery study correspond to. Do they mean that they tested 5 replicates of 5 samples fortified at 0.1 tested on different days; 5 replicates of 5 samples fortified at 1 on different days and 5 replicates of 5 samples fortified at 1 on different days? If so, why do you do it 5 times? and are the results of these 5 replicate 5-day samples used to calculate the interday recovery? On which organization or document did you base your approach to method validation?

Response: Thank you very much for your positive and constructive comments. Figure 1 illustrates an analytical batch used to evaluate the accuracy and precision of the UPLC method. One such analytical batch generates recoveries and intra-day relative standard deviation (RSD) for three quality control (QC) concentrations (0.1, 1, and 10 μg/g), while five such batches yield inter-day RSD for the same concentrations. The analytical accuracy and precision were evaluated according to European Commission Decision 2002/657/EC [1]. Five analytical batches were performed to fully evaluate the reproducibility of the method across different batches. We have rewritten the sentence regarding the evaluation of analytical accuracy and precision (please refer to lines 363-365) and cited the reference in the corresponding section of the revised manuscript (please refer to line 351 and line 500). We hope this change can make the expression clearer and our explanation meets your requirements.

Figure 1. an analytical batch for evaluating the precision and accuracy of the UPLC method

Reference

  1. European commission decision 2002/657/EC. Official Journal of the European Communities, 2002, 221, 8–36.

Reviewer 2 Report

Comments and Suggestions for Authors

I would like to thank the authors for making the changes and additions to the manuscript suggested by me. Thanks to this, the authors improved their manuscript, which can be published in this form.

Author Response

Thank you very much for your positive and constructive comments.